# Computerized False Belief Tasks Impact Mentalizing Ability in People with Williams Syndrome

**DOI:** 10.3390/brainsci13050722

**Published:** 2023-04-26

**Authors:** Ching-Fen Hsu, Shi-Yu Rao

**Affiliations:** School of Foreign Languages, Laboratory for Language Pathology and Developmental Neurosciences, Hunan University, Lushan Road (S), Yuelu District, Changsha 410082, China

**Keywords:** false belief, Williams syndrome, theory of mind, social cognition

## Abstract

People with Williams syndrome (WS) are characterized by hyper sociability, fluency in languages, and advantageous face-processing skills, leading to the proposal of a social module. Previous studies on the mentalizing abilities of people with WS using two-dimensional pictures, including normal-like, delayed, and deviant behaviors, have yielded mixed results. Thus, this study examined the mentalizing ability of people with WS through structured computerized animations of false belief tasks to investigate whether inferences about other people’s minds can be improved in this population. Participants were shown animations containing unexpected location and content changes. After viewing each animation, participants had to answer four types of questions relating to character identification, reality, memory, and false belief. Their responses were recorded and analyzed. A comprehension of false belief was observed in 4-year-old healthy children, whereas children with WS showed enhanced comprehension of false belief (until they attained a chronological age [CA] of 5.9 years), suggesting an improvement in the theory of mind resulting from viewing structured computerized animations. This age is earlier than that reported by previous studies for using theory of mind to pass false belief tests (CA 9 years), even challenging the age at which individuals failed to pass the tests (CA 17.11 years). Structured computerized animations enhanced the mentalizing ability of people with WS to a certain extent. Compared to the typically developing controls, people with WS presented with a lower developmental level in processing false belief tasks. This study has educational implications for the development of computerized social skills interventions for people with WS.

## 1. Introduction

Mentalizing other people’s minds is an important cognitive ability related to social cognition and interpersonal communication. It is realized through multiple aspects such as language, face processing, and joint attention. Premack and Woodruff [1] first proposed the theory of mind to account for the mentalizing ability in chimpanzees, and this phenomenon was subsequently described in humans by Wimmer and Perner [2]. The theory of mind, or mindreading, refers to the ability to understand other people’s mental states and to predict their behaviors. This mindreading ability has been investigated among people with neurodevelopmental disabilities, including those with Williams syndrome (WS). Investigating the mentalizing ability of other people’s minds in people with WS is important to have a broader understanding of their social cognition.

People with WS are a population with genetic deficits of chromosome 7q11.23. This syndrome is a rare disorder with a reported epidemiology of 1 in 7500 live births [3]. The syndrome results from missing genes in this region [4]. Consequently, people with WS are uneven in their cognition of relatively good language and poor visuospatial perception [5]. People with WS are characterized as having intellectual disability with an average of 55 IQs. They have advantageous lexical semantics, fluent expressive language, and good facial recognition. However, they find it difficult to build interpersonal relationships with peers and have impaired social cognition. To solve this conflict, an investigation of the social ability of people with WS is essential.

False belief tests are standard tests of the theory of mind, including the hallmark unexpected location change and unexpected content change tasks. The hallmark tests were developed by Baron-Cohen, Leslie, and Firth [6]. The hallmark unexpected location change task refers to the Sally-Anne task, aiming at probing the mentalizing ability of other people’s minds regarding a location change of the target objects. The task adheres to the following script: “Sally and Anne were playing marbles in a room. Then, Sally put the marbles in her basket and left the room. Anne was very naughty and took the marbles out of the basket and put them into her box. After a while, Sally came back into the room and wanted to play with her marble”. After listening to the story, participants were asked the false belief question (Where would Sally look for her marbles?) and the control questions (Where are the marbles now? [reality question], Where were the marbles in the beginning? [memory question]). The hallmark unexpected content change task refers to the Smarties test [7], aiming at probing the mentalizing ability of other people’s minds regarding a content change of target objects. The task displays a tube of smarties to children who are asked what the content inside the tube might be. Actually, there are no smarties inside the tube, rather there are pencils. Four-year-old children generally successfully passed these hallmark false belief tests. Since these tests were developed to probe the mentalizing ability of other people’s minds by mimicking one of the character’s minds, these tests are first-order tests of the theory of mind. In addressing the issue of theory of mind in people with developmental disabilities, all previous studies used traditional, two-dimensional testing materials. In this study, we used computerized animation videos as the testing materials to examine the possibility of the enhancement of the mentalizing ability of people with WS.

A previous study reported that people with WS (9–23 years) showed better mindreading ability in the first-order tests of the theory of mind than those with autism; however, the study lacked a control group [8]. In another study with 13 participants with WS (17–37 years) who completed a mindreading test using their eyes, people with WS performed better in inferring mental states than those with Prader-Willi syndrome (PWS; another population with genetic deficits on chromosome 15 at q11-13 region with even cognitive profiles of language and visuospatial abilities); however, they were worse than the healthy control groups [9]. These results suggest that people with WS are relatively good at mentalizing other people’s minds using the mindreading from the eyes test, but do not reach the developmental level of the healthy control groups. There is still a gap in the mindreading ability of people with WS when compared to individuals displaying typical development.

Children with WS exhibited increased impairment in mentalizing other people’s minds compared to children with PWS or non-specific mental retardation (NSMR) when performing false belief tasks of location and content change [10]. This finding suggests that a deficiency of the theory of mind in people with WS starts early in childhood. This view is comparable with the representational redescription model proposed by Karmiloff-Smith [11]. The ability to mentalize other people’s minds in people with WS results from a modularized process together with fluent language and social interaction given the innate tendency to, for example, look at human faces. However, early gene mutation has a devastating influence on later development in people with WS, as proposed in neuro constructivism [12].

In addition to the hallmark false belief tests, explanation of action assessments have been used to evaluate the ability of theory of mind in people with WS [10]. In Tager-Flusberg and Sullivan’s study, four types of stories probing desire, emotion, cognition, and causal reasoning were used to test three groups of participants (people with WS, PWS, and NSMR). The results showed that people with WS (CA 4.6–8.7 years) were no better than the other two groups at explaining human actions. Additionally, people with WS performed worse in tasks involving causal reasoning compared to those of cognition (the condition the other two groups struggled the most with). This finding suggests that people with WS were impaired in processing non-psychological or physical-related causal reasoning; however, their mentalizing ability was at the same level as that of people with PWS and NSMR. It should be noted, however, that the test stimuli were verbal narrations without visual pictures, which have been demonstrated to improve the ability of people with WS to integrate information because of the social information in the pictures [13].

Computer-based technology tools can enhance social skills in people with developmental disabilities and people with neural disorders through computer-assisted programs, virtual reality, and robotics [14]. In the study by Golan and Baron-Cohen [15], people with Asperger Syndrome (AS) and people with high functioning autism (HFA) were trained using computer-based programs to recognize the contexts of facial and voice emotions. This training was aimed at improving the socio-emotional abilities of these clinical individuals. Participants in the intervention group underwent a 2-h training every day for 10 to 15 weeks to recognize complex emotions such as insincerity and other mental states. Participants were evaluated before and after the intervention. Each participant was evaluated using several software programs, including the Cambridge Mind-Reading Face-Voice Battery, and psychometric analysis to recognize mental states using formats such as reading the mind in the eyes, voice, and films. Two control groups included people with either of the syndromes AS and HFA who did not receive the interventions and those with typical development. It was hypothesized that the two clinical groups would perform worse before the intervention than typical developers. Moreover, it was hypothesized that people with AS and people with HFA would improve after taking the training with software programs. These results confirmed the hypotheses. People with AS and people with HFA who received training using interactive, multimedia, and educational software improved in recognizing complex emotions and mental states, proving that computer-based interventions can help people with neurodevelopmental disabilities. Hopefully, people with WS should benefit from such training programs as well.

Social skills interventions have proven effective for people with neurodevelopmental disorders. Extant research has reported the possibility of using such interventions for people with autism by demonstrating a lack of significant difference between traditional face-to-face social skills training programs and behavioral cognitive intervention programs [16]. Fisher and Morin [17] developed interventions for people with WS by using the training programs of UCLA PEERs for Adolescent Programs manual [18], Health and Family Life Education Common Curriculum [19], and Think Social [20]. Before implementing these programs, parental questionnaires were distributed to understand the social skills of people with WS. Next, discussions were held with parents of adults with WS to confirm their social skill problems and to develop specific intervention programs for them (social skills training program for people with WS [SSTP-WS]). Pre- and post-tests of social skills interventions on people with WS were conducted with effective results observed within two days. Their study demonstrates that SSTP-WS is a promising intervention tool for people with WS. The acceptability, feasibility, and efficacy of this training program were further confirmed by an 8-week-long SSTP with people with WS [21]. Both studies demonstrated effective face-to-face telehealth social skills training in people with WS. However, there have not been any computerized intervention programs on improving the social cognition of people with WS. Before concluding the intervention effect, empirical studies demonstrating improvement in mentalizing the minds of people with WS should be conducted.

The aim of the current empirical study was to examine whether computerized animations could improve the mentalizing ability of people with WS. It was hypothesized that computer-based technology would have some impact on the cognitive behaviors of people with WS. Moreover, the effect of advanced technological research methods should be revealed. With these findings, interventions are possible in the future for people with WS.

## 2. Method

### 2.1. Participants

A total of 22 people with WS (mean CA = 9.9, SD = 3.1, 12F/10M, range = 5.9–18.1; mean MA = 6.4, SD = 2.4, range = 3.8–12.3) were recruited for the location-change false belief task; 17 people with WS (mean CA = 10.3, SD = 3.3, 8F/9M, range = 6.6–18.1; mean MA = 6.7, SD = 2.4, range = 4.0–12.3) were recruited for the content-change false belief task. All people with WS were diagnosed with missing genes on chromosome 7q11.23 in hospitals at various ages. They were recruited from the annual convention on health checking in the Children’s Hospital Zhejiang University School of Medicine (Hangzhou city) and provinces across mainland China such as Jiangsu (Shanghai city, Wuxi city) and Hebei (Beijing). Healthy controls were individually matched with people with WS based on their CA and MA using the Wechsler Scale of Intelligence for Children (Chinese version used in mainland China). The gender of each participant with WS and healthy control was also matched. No difference was observed in age between the CA or MA group and people with WS.

A total of 20 healthy 3- and 4-year-old children from four kindergartens in Changsha, in Hunan Province, China, were recruited in each group. The age difference between the groups was significant [3 years: mean age = 3.4, SD = 0.2; 4 years: mean age = 4.2, SD = 0.2; *t*(19) = 16.998, *p* < 0.001]. We aimed to verify the validity of the testing trials and to examine whether the transition from 3 to 4 years of age is critical in the Chinese education environment for children’s development pertaining to false beliefs. Standard false belief tasks with changes in location and content were conducted. The background information on all participants is listed in Table 1. This study was approved by the Institutional Review Board of the School of Foreign Languages of Hunan University, China. Before the experiment began, each participant’s guardian signed an informed consent form.

### 2.2. Materials and Design

A total of two false belief tasks were used: the unexpected location-change task and the unexpected content-change task. A total of 20 trials were conducted for each task. All trials were presented in the form of cartoon videos (length of the location task = 26.20 min, mean = 1.32, SD = 0.05; length of the content task = 26.83 min, mean = 1.35, SD = 0.07; total length of the two tasks = 53.03 min). Two additional practice trials were performed before the experiment began (length of the location task = 2.70 min, mean = 1.35, SD = 0.07; length of the content task = 2.82 min, mean = 1.42, SD = 0.02; total length of the two tasks = 5.52 min). Each trial comprised a scenario with two cartoon protagonists acting out a script.

Five pairs of cartoon characters were presented in the two tasks: Winnie the Pooh and Tigger, Mickey Mouse and Donald Duck, Pleasant Goat and Grey Wolf, Tom and Jerry, and SpongeBob and Patrick Star. These cartoon characters were selected from the most popular children’s films of the last 3 to 5 years in China to ensure that they were easily recognizable. As such, all of the participants were familiar with each protagonist. The names of all the chosen characters were of the same length when written in Chinese. The cartoon characters were purchased from a picture-producing company that owns their copyrights. The Appendix A animation videos were created by using Photoshop software and then the narrations were recorded using a cell phone. At the beginning of each location- and content-change video, two characters were introduced consecutively.

Each scenario followed a template with the sequence of a general setting, an action, a motivation, a confirmed motivation action, a character who leaves temporarily, key actions, a false belief-inducing action, and comprehension questions. An example of the unexpected location-change task is provided in Table 2. Each scenario was well-designed in its structure and details. In the parts relating to motivation and key actions, three movements were included. The crucial turning point was the key actions that might introduce false beliefs to participants. Each scenario was followed by comprehension questions. Each participant responded to all questions regarding their recognition of cartoon characters, memory, reality, and the false belief scenarios. No pair of cartoon characters were displayed consecutively. Participants’ responses to all trials were coded as 1 if correctness was met or 0 if incorrect responses were given. All experimental trials were new, creative, and different from previous studies in their computerized animation design and parallel structured design in the unexpected location-change task and unexpected content-change task. All the experimental trials were revised repeatedly to meet the need for testing the theory of mind on children. In this study, the false belief questions were experimental trials, and other non-false belief questions were control trials. The non-false belief questions included questions probing memory, reality, and character recognition. The testing materials in this study were computerized animation videos in both the experimental and control trials. In the data analyses, comparisons were conducted between the experimental trials and the control trials on accuracy and reaction times.

Several factors were considered when creating scenarios based on children’s developmental stages in language comprehension. Time expressions, for example, before, after, and then, were removed and replaced with non-referential time point terms, such as okay and at that time. Children were able to understand the sequence of actions upon watching the videos. Parallel video structures of the content-change task were created, as shown in Table 3. All participants received two practice trials before the experiment began. The computerized scenarios designed in this study have not been used before in any related tests.

### 2.3. Procedure

Participants were individually tested in a quiet room. Each participant watched videos with unexpected location-change scenarios or content-change scenarios. At the end of each scenario, each participant responded to the comprehension questions probing false belief and four control questions (the recognition of two characters, memory, and reality). The experimenter recorded participants’ responses on answer sheets simultaneously. Trials were presented randomly.

## 3. Results

The analysis plan of this study was to compare the performance of 3-year-old and 4-year-old children to ensure the validity of the Chinese testing materials in false belief tasks. Then, analyses were conducted in each false belief task in a multivariate model to examine the group effect on each type of comprehension question. Only correct responses to the experimental trials and control trials were analyzed. The comparisons of the experimental trials and the control trials were conducted using binominal nonparametric statistical analyses. Before analyzing the data of people with WS, we analyzed the results collected from 3-year-old and 4-year-old typically developing children to ensure whether the classical false belief test was valid across this age boundary in China. Non-parametric binominal statistical analysis was conducted. The average of all correct responses was calculated and the higher percentage of the type of observatory response was compared to the expected value of the chance level, 0.50. If the difference between an observatory value and the expected value reached significance, it could be concluded that the particular performance (e.g., false belief passing rate or failing rate) was valid. This validation would confirm that the participants in that group succeeded or failed the test.

### 3.1. Analyses of Healthy 3- and 4-Year-Old Controls

Correct responses, including accurate recognition of characters (character recognition questions), accurate inferences about false beliefs (false belief questions), accurate identification of the location or content of the targeted object in the final situation (reality question), and accurate indication of the original position of or content in the container (memory question), were analyzed.

Non-parametric binomial statistical tests were used to analyze the location-change and content-change task data. Both 3- and 4-year-old children passed the recognition of characters test (*p* < 0.001); both groups showed highly accurate percentage values (the location-change task, 100% in both age groups [in 3-year-olds, SD = 0.05; 4-year-olds, SD = 0]; the content-change task, 99% [SD = 0.07] in 3-year-olds and 100% [SD = 0] in 4-year-olds) in character comprehension in the videos. The memory question also reached high accuracy levels in both groups at *p* < 0.001 (the location-change task, 89% [SD = 0.31] in 3-year-olds, 100% [SD = 0] in 4-year-olds; the content-change task, 89% [SD = 0.31] in 3-year-olds, 99% [SD = 0.07] in 4-year-olds). Fisher’s exact tests showed a significant difference in the memory test between the 3- and 4-year-old groups in the location-change task (*p* = 0.00007) and the content-change task (*p* = 0.005). Both age groups responded to the reality question correctly at *p* < 0.001 (the location-change task, 3-year-olds, 90% [SD = 0.31], 4-year-olds, 100% [SD = 0]; the content-change task, 3-year-olds, 95% [SD = 0.23], 4-year-olds, 100% [SD = 0]). Fisher’s exact test showed a significant difference between the 3- and 4-year-old groups regarding the reality question in the location-change task (*p* = 0.0015); however, the difference was not significant between groups in the content-change task.

Regarding the false belief questions, 3-year-old children showed extremely low accuracy in the location-change task (5%, SD = 0.22) and in the content-change task (4%, SD = 0.20) at *p* < 0.001, whereas 4-year-old children showed a relatively high level of accuracy in responding to the false belief question of location (99%, SD = 0.10) and of content (98%, SD = 0.14) at *p* < 0.001. Multivariate analyses of variance revealed group differences in false belief tasks, *F*_(1,798)_ = 6142.09, *p* < 0.001, *ƞ*^2^ = 0.885, suggesting that 4-year-olds had attained the milestone of discerning false belief compared to 3-year-olds. Fisher’s exact test also showed a significant difference between the 3- and 4-year-old groups at *p* < 0.00001 regarding the false belief question in the location-change and content-change tasks. Group differences were observed along the reality [*F*_(1,798)_ = 46.81, *p* < 0.001, *ƞ*^2^ = 0.055] and memory [*F*_(1,798)_ = 48.06, *p* < 0.001, *ƞ*^2^ = 0.057] dimensions, indicating higher accuracy among 4-year-olds than 3-year-olds. The responding percentage of 3-year-old and 4-year-old children is graphed in Figure 1. Put together, these differences generally imply advanced cognitive development among older children.

### 3.2. Analyses of the Unexpected Location-Change Task

Multivariate analyses of variance were performed with the correct responses to each trial of each type. This was done in the unexpected location-change task of the within-participant factor groups and the between-participant factor groups. No differences emerged in the two questions regarding the recognition of the two characters across groups (100% in all groups), suggesting that the children clearly comprehended the tested animated videos. Significant differences were observed in responding to questions related to memory, reality, and false belief [memory, *F*_(2,1317)_ = 32.01, *p* < 0.001, *ƞ*^2^ = 0.46; reality, *F*_(2,1317)_ = 20.90, *p* < 0.001, *ƞ*^2^ = 0.031; false belief, *F*_(2,1317)_ = 175.86, *p* < 0.001, *ƞ*^2^ = 0.211]. The question of reality reached significance, as detected by the Tukey method [CA (100%, SD = 0) vs. MA (96%, SD = 0.187), *p* < 0.001; CA vs. WS (92%, SD = 0.278), *p* < 0.001; MA vs. WS, *p* < 0.001]. Another significant difference was observed for the question on memory [CA (100%, SD = 0) vs. MA (99%, SD = 0.082), *p* >0.05; CA vs. WS (92%, SD = 0.271), *p* < 0.001; MA vs. WS, *p* < 0.001]. Another difference was uncovered for the key question on false belief [CA (100%, SD = 0) vs. MA (91%, SD = 0.291), *p* < 0.001; CA vs. WS (60%, SD = 0.490), *p* < 0.001; MA vs. WS, *p* < 0.001]. Overall, people with WS showed the lowest accuracy, and the MA controls demonstrated mid-range accuracy values. This finding suggests that people with WS fared worse in mentalizing other people’s minds than people with MA.

A nonparametric binomial statistical test was used to analyze each group based on each type of question. The results revealed significant differences in comprehending questions on character recognition, reality, and memory [two character recognition questions: CA, 100%; MA, 100%; WS, 100%; memory question: CA, 100%; MA, 99%; WS, 92%; reality question: CA, 100%; MA, 96%; WS, 92%]. Significant differences also emerged in response to false beliefs between people with WS and the MA and CA controls at *p* < 0.001 [CA, 100%; MA, 91%; WS, 60%]. People with WS were accurate only 60% of the time regarding the question related to the location-change false belief task. The responding percentage of the three groups in the unexpected location-change task was graphed in Figure 2.

### 3.3. Analyses of the Unexpected Content-Change Task

Multivariate variance analyses with question types for the within-participants factor groups and for the between-participants factor groups were conducted. The results revealed no group differences in character recognition questions (all participants recognized cartoon characters correctly), but significant group differences in those pertaining to memory [*F*_(2,1017)_ = 5.35, *p* = 0.005, *ƞ*^2^ = 0.010], reality [*F*_(2,1017)_ = 3.02, *p* = 0.049, *ƞ*^2^ = 0.006], and false belief [*F*_(2,1017)_ = 197.09, *p* < 0.001, *ƞ*^2^ = 0.279]. Post-hoc analyses with the Least Significant Difference method revealed differences between the WS group (99%, SD = 0.094) and the CA (100%, SD = 0) and MA groups (97%, SD = 0.161) in their responses to reality questions (*p* = 0.034). No difference emerged between the CA and MA groups. Another significant difference in comparison using the Tukey method was observed from the distinct processing of the WS group (63%, SD = 0.483) and the control groups (CA, 100%, SD = 0; MA, 100%, SD = 0) in their response to false belief (*p* < 0.001). No difference was observed between the CA and MA groups. Concerning questions about memory, the Tukey method revealed that the difference between the CA and MA groups was significant (*p* = 0.004). No difference was observed between the WS group and the healthy controls.

Separate analyses of each group revealed significant differences in response to each type of question at *p* < 0.001. The pattern of response to false belief was reversed in the WS group. People with WS passed the false belief test 63% of the time (in contrast to the 100% passing rate of the CA and MA groups), suggesting that people with WS were impaired in the processing of false beliefs. The responding percentage of the three groups in the unexpected content-change task is graphed in Figure 3.

## 4. Discussion

This study examined the understanding of first-order false beliefs in people with WS. Structured animated video clips were used to test unexpected locations and content, revealing that people with WS made inferences through the differentiation of false belief from reality and memory by viewing the animations. In this study, the mean CA of participants with WS (60% in the location-change task, 63% in the content-change task) who passed the false belief tests was 5.9 years (range = 5.9–18.1) [MA 3.8 years (range = 3.8–12.3)] in the location-change task and 6.6 years (range = 6.6–18.1) [MA 6.7 years (range = 4.0–12.3)] in the content-change task. The passing CA of people with WS in this study was lower than the passing CAs in previous studies addressing the issue of the theory of mind. Specifically, Karmiloff-Smith et al.’s study [8] revealed a CA range of 9–23 years (MA, BPVS 5.3–10.7 years) and Tager-Flusberg’s study [9] revealed a CA range of 17.11–37.0 years (MA, Peabody Picture Vocabulary Test-Revised [PPVT-R] mean 12.10 years). Our results not only demonstrate the different processing passing rates of location-change and content-change false belief tasks, but also show that structured computerized animations enhance the mentalizing ability of people with WS. The number of people with WS who passed the false belief tasks was higher than of those who failed to do so; however, people with WS still showed the lowest accuracy among all groups. This finding confirms our hypothesis that technological tools can improve the performance in false belief tasks of people with WS [14].

The CA difference (i.e., 5.9 and 6.6 vs. 4) implies a discrepancy in processing between computerized three-dimension animation and traditional two-dimensional images. Though the mental ages in the study’s two tasks were close to the age for passing false belief tasks among those aged 3.8 and 4 years, the passing percentages were under 100 percent (60% in the location-change task, 63% in the content-change task). The cause of the CA difference might be due to a deficiency in integrating contextual information in people with WS [13,22,23,24,25,26,27,28,29,30,31,32,33,34], difficulty in understanding task demands [35], or superficial knowledge of lexical semantics [36]. This deficiency was evident in integrating word meaning into context during sentence processing [13,32], connecting words in a semantic organization [29,30], delayed performances on causal inference through comprehension of ambiguous words [24], deviant contextual integration using pictures [13], and deviant integration of propositions in people with WS [34]. Further atypical neurological information processing across verbal and nonverbal domains was reported in conceptual formation [22], semantic priming [29], and face recognition [33,37]. These deficits might have caused an impaired ability to mentalize other people’s minds in people with WS. Although computerized video animations did improve the mentalizing ability of people with WS, the deviant pattern in false belief revealed in the current study implies an atypical development of the theory of mind in this population. Hence, a relatively developed social module for people with WS is proposed.

This atypical development leads to deficient social cognition in people with WS, which is not at a level equivalent to that of CA-matched healthy controls [38]. Einfeld, Tonge, and Florio et al. [38] reported increased behavioral and emotional disturbance in people with WS aged 9.2 years when compared to populations with intellectual disabilities aged 12 years. People with WS were significantly over-affectionate, sensitive to anxiety, preoccupied with certain ideas, inappropriately happy or elated, wandering around, and continually repeating words. These emotional problems may result in the atypical social behaviors of people with WS and the deficient ability to mentalize other people’s minds. Furthermore, people with WS have been shown to be atypical in emotion recognition through narrations and replacement while aiming at targeted emotions [31,39]. Moreover, people with WS are delayed in their processing of anger and surprise emotions compared to the typically developing controls.

Language ability is an important factor that can influence the development of the theory of mind. Evidence from people with deafness and those with visual impairments showed delayed development of mentalizing abilities in both compared to typically developing controls due to the paucity of language input during the early stages of their lives [40]. Hence, people with WS were deficient in their ability to mentalize other people’s minds due to impaired language abilities tested through their understanding of false beliefs. However, in a prior study, language was shown to not play a role in the comprehension of other people’s minds in people with WS, as their language ability did not predict their mindreading ability in verbal and low-verbal false belief tasks [35].

Different patterns of social interactions with their parents, siblings, and other people early in the lives of people with WS might be a determining factor influencing their development of mentalizing other people’s minds and social interactions. Moreover, the atypical processing of faces may contribute to the deviant social cognition of people with WS [41]. Even the ability to understand narrations expressed in nouns or verbs has different contextual effects on people with WS [32]. However, executive functions evaluated by working memory and tapping tasks are unrelated to false belief performances in people with WS [9]. Although emotional cues help people with WS understand others’ minds better [42], future studies exploring false beliefs at the neurological level in people with WS are needed to lend support to the relatively developed social module. There are other research methods that could access the issue of theory of mind in people with WS, such as mindreading from eyes, emotional understanding through faces, and more complicated second-order false beliefs. This study took animation videos of first-order false belief tests as experimental materials to probe the ability of theory of mind in people with WS. This research method is creative, challenging, and interesting in the field of developmental disabilities. This contributes to the use of advanced computerized animations to improve the mindreading ability of people with WS and explore the possibility of educational interventions for this population in the future.

## Figures and Tables

**Figure 1 brainsci-13-00722-f001:**
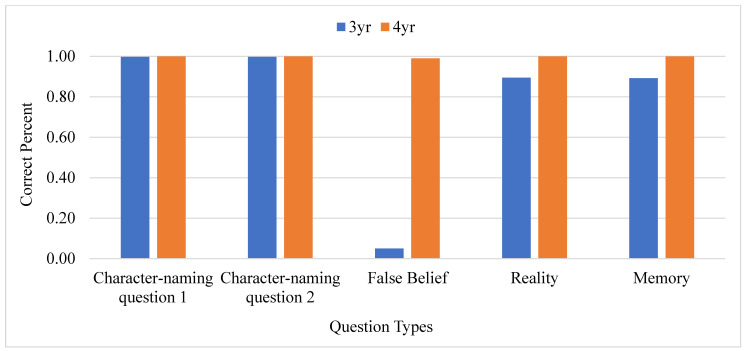
3-year-old and 4-year-old type percentage. Note: 3 yr represents 3-year-old children; 4 yr represents 4-year-old children.

**Figure 2 brainsci-13-00722-f002:**
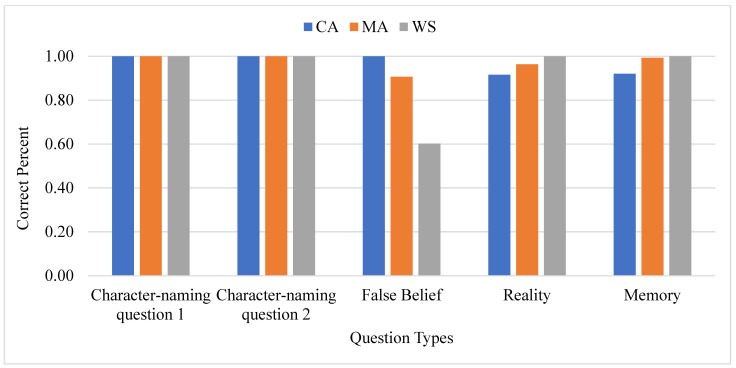
Three groups type percentage- location change.

**Figure 3 brainsci-13-00722-f003:**
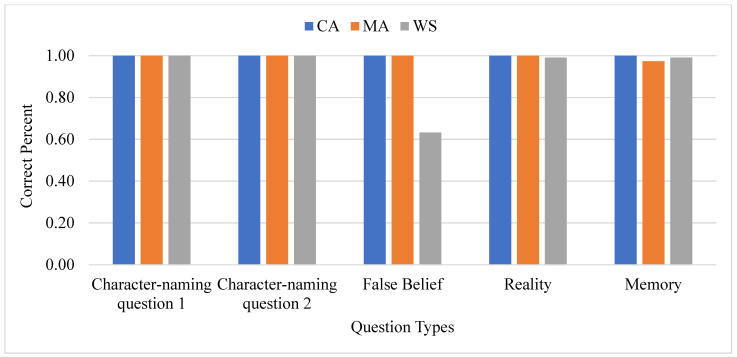
Three groups type percentage-content change.

**Table 1 brainsci-13-00722-t001:** Background information of participants.

Task	Group	N	F:M	Mean CA (SD)	Range	Mean MA (SD)	Range	Statistical Results
Unexpected Location Task	CA	22	12:10	9.9 (3.3)	5.7–18.7			CA vs. MA, *t*(21) = 10.080, *p* < 0.001 CA vs. WS-CA, *t*(21) = −0.084, *p* = 0.934 CA vs. WS-MA, *t*(21) = 10.560, *p* < 0.001
	MA	22	12:10	6.3 (2.3)	3.8–12.2			MA vs. WS-CA, *t*(21) = −11.139, *p* < 0.001 MA vs. WS-MA, *t*(21) = −0.902, *p* = 0.377
	WS	22	12:10	9.9 (3.1)	5.9–18.1	6.4 (2.4)	3.8–12.3	
	3 years old	20	10:10	3.4 (0.2)	3.0–3.6			3 years old vs. 4 years old, *t*(19) = 16.998, *p* < 0.001
	4 years old	20	10:10	4.2 (0.2)	4.0–4.6			
Unexpected Content Task	CA	17	8:9	10.4 (3.5)	6.3–18.7			CA vs. MA, *t*(16) = 8.674, *p* < 0.001 CA vs. WS-CA, *t*(16) = 0.086, *p* = 0.933 CA vs. WS-MA, *t*(16) = 5.555, *p* < 0.001
	MA	17	8:9	6.6 (2.4)	4.0–12.2			MA vs. WS-CA, *t*(16) = −5.567, *p* < 0.001 MA vs. WS-MA, *t*(16) = −0.108, *p* = 0.916
	WS	17	8:9	10.3 (3.3)	6.6–18.1	6.7 (2.4)	4.0–12.3	
	3 years old	20	10:10	3.4 (0.2)	3.0–3.6			3 years old vs. 4 years old, *t*(19) = 6.998, *p* < 0.001
	4 years old	20	10:10	4.2 (0.2)	4.0–4.6			

Note: F:M refers to the ratio of female to male participants; CA, chronological age; MA, mental age; WS, Williams syndrome; SD, standard deviation.

**Table 2 brainsci-13-00722-t002:** Example of the unexpected location-change task (with original Chinese text).

Structure	Contexts
General Setting	唐老鸭和米老鼠一起坐在阳台上晒太阳。 Donald Duck and Mickey Mouse sat on the balcony and enjoyed a sunbath.
Action	唐老鸭把花放进篮子里。 Donald Duck put the flowers in the basket.
Motivation (three actions)	唐老鸭和米老鼠坐了一会，唐老鸭觉得有点渴，想去喝水。 Donald Duck and Mickey Mouse sat for a while (verb 1). Donald Duck was thirsty (verb 2) and went to drink water (verb 3).
Confirmed Motivation Action	唐老鸭离开阳台，喝水去了。 Donald Duck left the balcony and went to drink water.
Left Character	这时，阳台上只剩下米老鼠。 At this time, only Mickey Mouse was left on the balcony.
Key Actions (one setting + three actions)	米老鼠很调皮，把花从篮子里拿出来，放进柜子里，再关上柜门。 Mickey Mouse was very naughty. He took the flowers out of the basket, put them in the cabinet, and then closed the door.
False Belief-Inducing Setting	过了一会，唐老鸭回到阳台，想闻闻花香。 After a while, Donald Duck returned to the balcony and wanted to smell the flowers.
Attention Arousing Greeting	好，小朋友， OK, dear,
Recognition Question 1	你知道哪个是唐老鸭？ Do you know which character Donald Duck is?
Recognition Question 2	你知道哪个是米老鼠？ Do you know which character Mickey Mouse is?
Belief Question	唐老鸭喝完水，回到阳台，唐老鸭会去哪里找花？ Donald Duck finished drinking water and went back to the balcony. Where would Donald Duck look for flowers?
Reality Question	现在花在哪里？ Where are the flowers now?
Memory Question	一开始唐老鸭把花放在哪里？ Where did Donald Duck put the flowers at first?

**Table 3 brainsci-13-00722-t003:** Example of the unexpected content-change task (with original Chinese text).

Structure	Contexts
General Setting	喜羊羊和灰太狼一起来到图书馆。 A pleasant goat and grey wolf came to the library together.
Action	他们在图书馆里准备看书。 They are ready to read the books in the library.
Motivation (three actions)	翻开书，喜羊羊和灰太狼有点看不清，想找副眼镜。 Opening the books, the pleasant goat and grey wolf could not see clearly.
Confirmed Motivation Action	于是，喜羊羊离开图书馆，去找眼镜。 So, the pleasant goat left the library to look for eyeglasses.
Left Character	这时，图书馆里只剩下灰太狼。 At this time, only the grey wolf was left in the library.
Key Actions (one setting; three actions)	喜羊羊回到图书馆，把眼镜盒放到灰太狼面前。这时候灰太狼要去拿书包，灰太狼离开了图书馆。 The pleasant goat returned to the library and put the eyeglasses box in front of the the grey wolf. Meanwhile, the grey wolf was going to get his school bag. Grey wolf left the library.
False Belief-Inducing Setting	哇，喜羊羊真调皮，居然把饼干装在眼镜盒里。 Wow, the pleasant goat was so naughty that he put cookies in his eyeglasses box.
Attention Arousing Greeting	好，小朋友， OK, dear,
Recognition Question 1	你知道哪个是喜羊羊？ Do you know which character the pleasant goat is?
Recognition Question 2	你知道哪个是灰太狼？ Do you know which character the grey wolf is?
Belief Question	灰太狼还没有打开过眼镜盒，灰太狼觉得眼镜盒里装的是什么？ The grey wolf has not opened the eyeglasses box yet. What does the grey wolf think is in the eyeglasses box?
Reality Question	现在你知道眼镜盒里装的是什么？ What is in the eyeglasses box now?
Memory Question	一开始喜羊羊去拿的眼镜盒里装的是什么？ What was in the eyeglasses box when pleasant goat went to get it at first?

## Data Availability

Data are available on request.

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
