# Peer review of "Computerized False Belief Tasks Impact Mentalizing Ability in People with Williams Syndrome"

_brainsci, 2023, doi:10.3390/brainsci13050722_

Round 1

Reviewer 1 Report

I understand that Ws is a small population however considering developmental changes your sample size is very small hence any conclusions can be very tentative. Could you please revise your discussion and provide more evidence about developmental changes, chosen age range and also why your sample size is underpowered? 

The literature review needs to be updated and provide a more critical approach as ToM is nowadays debatable. Please expand this section.

Author Response

Review #1

I understand that Ws is a small population however considering developmental changes your sample size is very small hence any conclusions can be very tentative. Could you please revise your discussion and provide more evidence about developmental changes, chosen age range and also why your sample size is underpowered? 

Answer: Since people with WS are rare disorder with epistemology of 1 in 7500 live births, it is very difficult to find the participants. In the unexpected location-change task, there were 20 participants, and in the unexpected content-change task, there were 17 participants. The numbers of participants were the average numbers observed in studies of developmental disabilities. These numbers are also very close to the healthy controls of 3-year-old children and 4-year-old children. The conclusion still has statistical power. In the section of discussion, the passing age in comprehending the false belief question in the location-change task was 5.9 years old, which was much closer to the critical age of 4 years old in passing the false belief tasks. The passing age of 5.9 years old in this study was much younger than the ages (8.5 years old, 17 years old) in the previous studies. This was all described in the section of discussion as following: “This study examined the understanding of first-order false beliefs in people with WS. Structured animated video clips were used to test unexpected locations and content, revealing that people with WS made inferences through the differentiation of false belief from reality and memory by viewing the animations. In this study, the mean age of participants with WS (60% in the location-change task, 63% in the content-change task) who passed the false belief tests was 5.9 years old (range = 5.9–18.1) in the location-change task and 6.6 years old (range = 6.6–18.1) in the content-change task. These ages were still over but closer to the critical age of 4 years old in passing the false belief tasks. The passing ages of people with WS in this study were much lower than the passing ages in the previous studies addressing the issue of the theory of mind (at least 8.5 years old in Karmiloff-Smith et al.’s study [8] and 17 years old in Tager-Flusberg’s study [9]. Our results not only demonstrate the different processing passing rates of location-change and content-change false belief tasks but also show that structured computerized animations enhance the mentalizing ability of people with WS. The number of people with WS who passed the false belief tasks was higher than of those who failed to do so; however, people with WS still showed the lowest accuracy among all groups. This finding confirms our hypothesis that technological tools can improve the performance in false belief tasks of people with WS [14].”

The reasons that might cause the discrepancy between the passing ages of this study and the previous studies were addressed in the following paragraph repeated here: “The age difference (i.e., 5.9 and 6.6 vs. 4) implies a discrepancy in processing between computerized three-dimension animation and traditional two-dimensional images. The cause of the difference might be due to a deficiency in integrating contextual information in people with WS [13, 22-34], difficulty in understanding task demands [35], or superficial knowledge of lexical semantics [36]. This deficiency was evident in integrating word meaning into context during sentence processing [13, 32], connecting words in a semantic organization [29, 30], delayed performances on causal inference through comprehension of ambiguous words [24], deviant contextual integration using pictures [13], and deviant integration of propositions in people with WS [34]. Further atypical neurological information processing across verbal and nonverbal domains was reported in conceptual formation [22], semantic priming [29], and face recognition [33, 37]. These deficits might have caused an impaired ability to mentalize other people’s minds in people with WS. Although computerized video animations did improve the mentalizing ability of people with WS, the deviant pattern in false belief revealed in the current study implies an atypical development of the theory of mind in this population. Hence, a relatively developed social module for people with WS is proposed.”

The literature review needs to be updated and provide a more critical approach as ToM is nowadays debatable. Please expand this section.

Answer: In the revising version of the paper, this section has been expanded. Two paragraphs were added. One paragraph introduced the definition of people with WS, and the other paragraph explained the first-order false belief tasks, i.e., the unexpected location-change task and the unexpected content-change task. These two paragraphs make the paper more comprehensible. There are nine paragraphs in the introduction in the revising version of the paper. The rationale and logic is very clear. The rationale of the revising paper was in nine paragraphs. In the first paragraph, the definition of metalizing ability or theory of mind was given with the rationale why this study was conducted on people with WS. In the second paragraph, the introduction of people with WS was mentioned to make the study more comprehensible to readers. In the third paragraph, the definition of the two false belief tasks were introduced. So, the readers may know what first-order false belief means and what the term hallmark refers to. In the fourth paragraph, the research gap in mentalizing other people’s minds was addressed. In the fifth paragraph, it was mentioned that people with WS may have impairments on mentalizing ability at early childhood as some previous studies revealed. This finding was confirmed by neuroconstructivism theory. In the sixth paragraph, it was proposed that visual stimuli might help people with WS in explaining other people’s minds rather than verbal narrations. This proposal is very important to address the key point of this study that computerized animations might improve mentalizing ability of people with WS. In the seventh paragraph, a brief introduction on the intervention effect of computer-based technology tools was addressed. In the eighth paragraph, a successful social skill training program on people with WS was introduced. In the ninth paragraph, the research question of this study was addressed. It was interested to examine whether visually presented computerized animations would improve the mentalizing ability of people with WS.

Reviewer 2 Report

Thank you for the opportunity to review “Do Computerized False Belief Tasks Impact Mentalizing Ability in People with Williams Syndrome?” This paper was interesting conceptually, but included a number of flaws. The below is a summary of feedback for improvement. Should the authors choose to resubmit, it is recommended that they adjust the manuscript accordingly.

The introduction requires more coherent organization and a clearer rationale for engaging in the present study. Currently, there is no definition of Williams Syndrome nor why it is important to examine theory of mind in this population. The introduction appears to skip around between ideas such as theory of mind, interventions, and theoretical underpinnings of Theory of Mind tasks without linking these ideas together.

The methodology and results require more information to determine whether matching procedures were appropriate, a clearer definition of the key tasks in the study, and why well-known cartoon characters were used. At present, it appears that this is a violation of intellectual property to use such characters, leading to ethical concerns over how these materials were developed. Further, there is no control task (i.e., non-animated ToM tasks). Without this task, it is an overreach to suggest that animated tasks are better at measuring ToM tasks than non-animated tasks.

No rationale is included for the analysis plan, which would help the reader understand those procedures (e.g., why is a multivariate analysis included and what variables were in the model?). Finally, the df error values are quite high (>1000), but the sample size is ~20 per group, leading this reviewer to have concerns about statistical procedures.

Author Response

Review Comments #2

Thank you for the opportunity to review “Do Computerized False Belief Tasks Impact Mentalizing Ability in People with Williams Syndrome?” This paper was interesting conceptually, but included a number of flaws. The below is a summary of feedback for improvement. Should the authors choose to resubmit, it is recommended that they adjust the manuscript accordingly.

The introduction requires more coherent organization and a clearer rationale for engaging in the present study.

Answer: The rationale of the revising paper was in nine paragraphs. In the first paragraph, the definition of metalizing ability or theory of mind was given with the rationale why this study was conducted on people with WS. In the second paragraph, the introduction of people with WS was mentioned to make the study more comprehensible to readers. In the third paragraph, the definition of the two false belief tasks were introduced. So, the readers may know what first-order false belief means and what the term hallmark refers to. In the fourth paragraph, the research gap in mentalizing other people’s minds was addressed. In the fifth paragraph, it was mentioned that people with WS may have impairments on mentalizing ability at early childhood as some previous studies revealed. This finding was confirmed by neuroconstructivism theory. In the sixth paragraph, it was proposed that visual stimuli might help people with WS in explaining other people’s minds rather than verbal narrations. This proposal is very important to address the key point of this study that computerized animations might improve mentalizing ability of people with WS. In the seventh paragraph, a brief introduction on the intervention effect of computer-based technology tools was addressed. In the eighth paragraph, a successful social skill training program on people with WS was introduced. In the ninth paragraph, the research question of this study was addressed. It was interested to examine whether visually presented computerized animations would improve the mentalizing ability of people with WS. The rationale of this study is now clear and logic.

Currently, there is no definition of Williams Syndrome nor why it is important to examine theory of mind in this population.

Answer: A paragraph mentioning about the definition of people with Williams syndrome was added as following: “People with WS are a population with genetic deficits of chromosome 7q11.23. This syndrome is a rare disorder with a reported epidemiology of 1 in 7500 live births [3]. The syndrome results from missing genes in this region [4]. Consequently, people with WS are uneven in their cognition of relatively good language and poor visuospatial perception [5]. People with WS are mentally retarded with an average of 55 IQs. They have advantageous lexical semantics, fluent expressive language, and good facial recognition. However, they find it difficult to build interpersonal relationships with peers and have impaired social cognition. To solve this conflict, an investigation of the social ability of people with WS is essential.”

The introduction appears to skip around between ideas such as theory of mind, interventions, and theoretical underpinnings of Theory of Mind tasks without linking these ideas together.

Answer: This question is related to the first one. In the revising version of this paper, the linkage between paragraphs in the introduction is clear and logic (see above). There are many different ways to probe the issue of the theory of mind. When talking about people with developmental disabilities, interventions are important in improving their ability of the theory of mind. This study addressed the possibility by using visualized computerized animations to enhance the mentalizing ability of people with WS, a population with genetic deficits. There are nine paragraphs in the introduction and the linkage logic between paragraphs is clear.

The methodology and results require more information to determine whether matching procedures were appropriate, a clearer definition of the key tasks in the study, and why well-known cartoon characters were used.

Answer: A paragraph in the section of introduction was added right after the definition of people with Williams syndrome about the introduction of the false belief tasks (key tasks) in this study on the unexpected location-change task and the unexpected content-change task as following: “False belief tests are standard tests of the theory of mind, including the hallmark unexpected location change and unexpected content change tasks. The hallmark tests were developed by Baron-Cohen, Leslie, and Firth [6]. The hallmark unexpected location change task refers to the Sally-Anne task, aiming at probing the mentalizing ability of other people’s minds regarding a location change of the target objects. The task adheres to the following script: ”Sally and Anne were playing marbles in a room. Then, Sally put the marbles in her basket and left the room. Anne was very naughty and took the marbles out of the basket and put them into her box. After a while, Sally came back into the room and wanted to play with her marble.” After listening to the story, participants were asked the false belief question (Where would Sally look for her marbles?) and the control questions (Where are the marbles now? [reality question], Where were the marbles in the beginning? [memory question]). The hallmark unexpected content change task refers to the Smarties test [7], aiming at probing the mentalizing ability of other people’s minds regarding a content change of target objects. The task displays a tube of smarties to children who are asked what the content inside the tube might be. Actually, there are no smarties inside the tube, rather there are pencils. Four-year-old children generally successfully passed these hallmark false belief tests. Since these tests were developed to probe the mentalizing ability of other people’s minds by mimicking one of the character’s minds, these tests are first-order tests of the theory of mind.”

Meanwhile, the reason why well-known cartoon characters were chosen was added in the section of materials and design as following: “Five pairs of cartoon characters were presented in the two tasks: Winnie the Pooh and Tigger, Mickey Mouse and Donald Duck, Pleasant Goat and Grey Wolf, Tom and Jerry, and SpongeBob and Patrick Star. These cartoon characters were selected from the most popular children’s films of the last 3 to 5 years in China to ensure they were easily recognizable. As such, all of the participants were familiar with each protagonist. The names of all the chosen characters were of the same length when written in Chinese. The cartoon characters were purchased from a picture-producing company that owns their copyrights. The animation videos were created by using Photoshop software and then the narrations were recorded using a cell phone. At the beginning of each location- and content-change video, two characters were introduced consecutively.”

At present, it appears that this is a violation of intellectual property to use such characters, leading to ethical concerns over how these materials were developed.

Answer: No violation of intellectual property in using these cartoon characters. To avoid this concern, explanation was added in the paragraph of materials and design as following: “Five pairs of cartoon characters were presented in the two tasks: Winnie the Pooh and Tigger, Mickey Mouse and Donald Duck, Pleasant Goat and Grey Wolf, Tom and Jerry, and SpongeBob and Patrick Star. These cartoon characters were selected from the most popular children’s films of the last 3 to 5 years in China to ensure they were easily recognizable. As such, all of the participants were familiar with each protagonist. The names of all the chosen characters were of the same length when written in Chinese. The cartoon characters were purchased from a picture-producing company that owns their copyrights. The animation videos were created by using Photoshop software and then the narrations were recorded using a cell phone. At the beginning of each location- and content-change video, two characters were introduced consecutively.”

Further, there is no control task (i.e., non-animated ToM tasks). Without this task, it is an overreach to suggest that animated tasks are better at measuring ToM tasks than non-animated tasks.

Answer: The research design in embedded in the animation videos with tests on comprehension questions, including the key false belief questions. The key false belief questions were compared to the control questions (memory questions, reality questions). The control questions were already there. Hence, there was no need to test other control task. Besides, it’s impossible to test 3-year-old and 4-year-old little children in such a long time (almost one hour in a task). The false belief task is well-design research method as a hallmark in the research field of developmental disabilities. The conclusion is trustworthy.

No rationale is included for the analysis plan, which would help the reader understand those procedures (e.g., why is a multivariate analysis included and what variables were in the model?).

Answer: A paragraph was added under the section of results to explain the analysis plan as following: “The analysis plan of this study was to compare the performance of 3-year-old and 4-year-old children to ensure the validity of the Chinese testing materials in false belief tasks. Then, analyses were conducted in each false belief task in a multivariate model to examine the group effect on each type of comprehension question.”

Finally, the df error values are quite high (>1000), but the sample size is ~20 per group, leading this reviewer to have concerns about statistical procedures.

Answer: The df error term in the unexpected location-change task was 1317, and the df error term in the unexpected content-change task was 1017. The statistical procedure was the same in the two tasks. A multivariate model was applied with the values from the five comprehension questions as the dependent variables and groups as the between-participant factor. This procedure was correct and the df error term was generated from the procedure in SPSS version 26.

Reviewer 3 Report

Review Report –

Brain Sciences

Do computerized false belief tasks impact mentalizing ability in people with Williams Syndrome?

Abstract

·         Lines from 18 – 20: a bit hard to follow and understand. Perhaps rephrase for better clarity and logic.

Introduction

·         Some more explanation for the tests of theory of mind would be helpful. For example, not all researchers are familiar with what it means by ‘first-order tests of theory of mind’ (line 38), ‘hallmark false belief tests’ (line 62).

·         More literature review on ‘advanced technological research method’ ‘computer-based technology’ in the field of the mentalizing ability would be helpful. Otherwise, it reads quite abrupt from the last paragraph of the introduction directly into the methods section.

·         Perhaps indicate your research questions toward the end of your introduction?

Method

·         How were the people with WS recruited? Where? Etc.

·         The Wechsler Scale of Intelligence for Children – is it a Chinese version? Please specify.

·         For Table 1, please also show the statistical test results that indicate the equivalence among different groups in terms of these key demographic variables.

·         It is recommended that the authors could at least briefly explain the ‘unexpected location-change task’ and ‘the unexpected content-change task’ a bit earlier in the methods section. Hard to imagine without much explanation.

·          Any online link to the experimental trials in the form of cartoon videos?

·         Who developed the experimental trials? Perhaps need a table to explain the experimental trials a bit?

·         It might be helpful to use some Figures to present some key results. Too much information to follow at its current format.

Discussion

·         More explanation regarding how the animation-based programs can actually help children with WS or other developmental disabilities to develop better mentalizing skills in reality is needed. 

Author Response

Review Comments #3

Abstract

  • Lines from 18 – 20: a bit hard to follow and understand. Perhaps rephrase for better clarity and logic.

Answer: The lines were revised as following: “These results suggest that people with WS are relatively good at mentalizing other people’s minds using the mindreading from the eyes test, but do not reach the developmental level of the healthy control groups. There is still a gap in the mindreading ability of people with WS when compared to individuals displaying typical development.”

Introduction

  • Some more explanation for the tests of theory of mind would be helpful. For example, not all researchers are familiar with what it means by ‘first-order tests of theory of mind’ (line 38), ‘hallmark false belief tests’ (line 62).

Answer: The term first-order tests of theory of mind and the meaning of the hallmark false belief tests were explained after explanation of the two false belief tests (unexpected location change task, unexpected content change task). The revised paragraph was added as following: “False belief tests are standard tests of the theory of mind, including the hallmark unexpected location change and unexpected content change tasks. The hallmark tests were developed by Baron-Cohen, Leslie, and Firth [6]. The hallmark unexpected location change task refers to the Sally-Anne task, aiming at probing the mentalizing ability of other people’s minds regarding a location change of the target objects. The task adheres to the following script: ”Sally and Anne were playing marbles in a room. Then, Sally put the marbles in her basket and left the room. Anne was very naughty and took the marbles out of the basket and put them into her box. After a while, Sally came back into the room and wanted to play with her marble.” After listening to the story, participants were asked the false belief question (Where would Sally look for her marbles?) and the control questions (Where are the marbles now? [reality question], Where were the marbles in the beginning? [memory question]). The hallmark unexpected content change task refers to the Smarties test [7], aiming at probing the mentalizing ability of other people’s minds regarding a content change of target objects. The task displays a tube of smarties to children who are asked what the content inside the tube might be. Actually, there are no smarties inside the tube, rather there are pencils. Four-year-old children generally successfully passed these hallmark false belief tests. Since these tests were developed to probe the mentalizing ability of other people’s minds by mimicking one of the character’s minds, these tests are first-order tests of the theory of mind.”

         More literature review on ‘advanced technological research method’ ‘computer-based technology’ in the field of the mentalizing ability would be helpful. Otherwise, it reads quite abrupt from the last paragraph of the introduction directly into the methods section.

Answer: A paragraph regarding computer-based technology tools used in intervention on people with developmental disabilities was added in the seventh paragraph. It was repeated in the following: “Computer-based technology tools can enhance social skills in people with developmental disabilities and people with neural disorders through computer-assisted programs, virtual reality, and robotics [14]. In the study by Golan and Baron-Cohen [15], people with Asperger Syndrome (AS) and people with high functioning autism (HFA) were trained using computer-based programs to recognize the contexts of facial and voice emotions. This training was aimed at improving the socio-emotional abilities of these clinical individuals. Participants in the intervention group underwent a 2-hour training every day for 10 to 15 weeks to recognize complex emotions such as insincerity and other mental states. Participants were evaluated before and after the intervention. Each participant was evaluated using several software programs, including the Cambridge Mind-Reading Face-Voice Battery, and psychometric analysis to recognize mental states using formats such as reading the mind in the eyes, voice, and films. Two control groups included people with either of the syndromes AS and HFA who did not receive the interventions and those with typical development. It was hypothesized that the two clinical groups would perform worse before the intervention than typical developers. Moreover, it was hypothesized that, people with AS and people with HFA would improve after taking the training with software programs. These results confirmed the hypotheses. People with AS and people with HFA who received training using interactive, multimedia, and educational software improved in recognizing complex emotions and mental states, proving that computer-based interventions can help people with neurodevelopmental disabilities. Hopefully, people with WS should benefit from such training programs as well.”

  • Perhaps indicate your research questions toward the end of your introduction?

Answer: The research question of the study was added in the end of introduction as following: “The current study examined whether computerized animations could improve the mentalizing ability of people with WS. It was hypothesized that computer-based technology would have some impact on the cognitive behaviors of people with WS. Moreover, the effect of advanced technological research methods should be revealed.”

Method

  • How were the people with WS recruited? Where? Etc.

Answer: The info was added in the section of participants: “Twenty-two people with WS (mean CA = 9.9, SD = 3.1, 12F/10M, range = 5.9–18.1; mean MA = 6.4, SD = 2.4, range = 3.8–12.3) were recruited for the location-change false belief task; 17 people with WS (mean CA = 10.3, SD = 3.3, 8F/9M, range = 6.6–18.1; mean MA = 6.7, SD = 2.4, range = 4.0–12.3) were recruited for the content-change false belief task. All people with WS were diagnosed with missing genes on chromosome 7q11.23 in hospitals at various ages. They were recruited from the annual convention on health checking in the Children’s Hospital Zhejiang University School of Medicine (Hangzhou city) and provinces across mainland China such as Jiangsu (Shanghai city, Wuxi city), and Hebei (Beijing). Healthy controls were individually matched with people with WS based on their CA and MA using the Wechsler Scale of Intelligence for Children (The Chinese version was used in mainland China). The gender of each participant with WS and healthy control was also matched. No difference was observed in age between the CA or MA group and people with WS.”

  • The Wechsler Scale of Intelligence for Children – is it a Chinese version? Please specify.

Answer: The info was added in the section of participants: “Twenty-two people with WS (mean CA = 9.9, SD = 3.1, 12F/10M, range = 5.9–18.1; mean MA = 6.4, SD = 2.4, range = 3.8–12.3) were recruited for the location-change false belief task; 17 people with WS (mean CA = 10.3, SD = 3.3, 8F/9M, range = 6.6–18.1; mean MA = 6.7, SD = 2.4, range = 4.0–12.3) were recruited for the content-change false belief task. All people with WS were diagnosed with missing genes on chromosome 7q11.23 in hospitals at various ages. They were recruited from the annual convention on health checking in the Children’s Hospital Zhejiang University School of Medicine (Hangzhou city) and provinces across mainland China such as Jiangsu (Shanghai city, Wuxi city), and Hebei (Beijing). Healthy controls were individually matched with people with WS based on their CA and MA using the Wechsler Scale of Intelligence for Children (The Chinese version was used in mainland China). The gender of each participant with WS and healthy control was also matched. No difference was observed in age between the CA or MA group and people with WS.”

  • For Table 1, please also show the statistical test results that indicate the equivalence among different groups in terms of these key demographic variables.

Answer: Table 1 has been revised with statistical values of age differences among groups. Please see the attachment.

  • It is recommended that the authors could at least briefly explain the ‘unexpected location-change task’ and ‘the unexpected content-change task’ a bit earlier in the methods section. Hard to imagine without much explanation.

Answer: The explanations of the unexpected location-change task and the unexpected content-change task were added in the second paragraph of introduction as following: “False belief tests are standard tests of the theory of mind, including the hallmark unexpected location change and unexpected content change tasks. The hallmark tests were developed by Baron-Cohen, Leslie, and Firth [6]. The hallmark unexpected location change task refers to the Sally-Anne task, aiming at probing the mentalizing ability of other people’s minds regarding a location change of the target objects. The task adheres to the following script: ”Sally and Anne were playing marbles in a room. Then, Sally put the marbles in her basket and left the room. Anne was very naughty and took the marbles out of the basket and put them into her box. After a while, Sally came back into the room and wanted to play with her marble.” After listening to the story, participants were asked the false belief question (Where would Sally look for her marbles?) and the control questions (Where are the marbles now? [reality question], Where were the marbles in the beginning? [memory question]). The hallmark unexpected content change task refers to the Smarties test [7], aiming at probing the mentalizing ability of other people’s minds regarding a content change of target objects. The task displays a tube of smarties to children who are asked what the content inside the tube might be. Actually, there are no smarties inside the tube, rather there are pencils. Four-year-old children generally successfully passed these hallmark false belief tests. Since these tests were developed to probe the mentalizing ability of other people’s minds by mimicking one of the character’s minds, these tests are first-order tests of the theory of mind.”

  • Any online link to the experimental trials in the form of cartoon videos?

Answer: Yes, the link were added in the third paragraph of materials and design as following: “Each scenario followed a template with the sequence of a general setting, an action, a motivation, a confirmed motivation action, a character who leaves temporarily, key actions, a false belief-inducing action, and comprehension questions. An example of the unexpected location-change task is provided in Table 2. Each scenario was well-designed in its structure and details. In the parts relating to motivation and key actions, three movements were included. The crucial turning point was the key actions that might introduce false beliefs to participants. Each scenario was followed by comprehension questions. Each participant responded to all questions regarding their recognition of cartoon characters, memory, reality, and the false belief scenarios. No pair of cartoon characters were displayed consecutively. All experimental trials were new, creative, and different from previous studies in their computerized animation design and parallel structured design in the unexpected location-change task and unexpected content-change task. All the experimental trials were revised repeatedly to meet the need for testing the theory of mind on children. The computerized animation clips were accessed through zenodo online at https://zenodo.org/record/7726825#.ZCVXn-gzaUk (DOI: 10.5281/zenodo.7726825).”

  • Who developed the experimental trials? Perhaps need a table to explain the experimental trials a bit?

Answer: The experimental design was from the first author, and video-making job was from the second author. All experimental trials were new, creative, and different from previous studies in its computerized animation design and parallel structured design in the unexpected location change task and unexpected content change task. All the experimental trials were revised repeatedly to meet the need for testing theory of mind on children. There are two tables (Table 2, Table 3) with examples from each task (the unexpected location change task, the unexpected content change task). In each table, the rationale and guiding steps to induce false belief were listed, respectively. The paragraph introducing the experimental trials was revised as following: “Each scenario followed a template with the sequence of a general setting, an action, a motivation, a confirmed motivation action, a character who leaves temporarily, key actions, a false belief-inducing action, and comprehension questions. An example of the unexpected location-change task is provided in Table 2. Each scenario was well-designed in its structure and details. In the parts relating to motivation and key actions, three movements were included. The crucial turning point was the key actions that might introduce false beliefs to participants. Each scenario was followed by comprehension questions. Each participant responded to all questions regarding their recognition of cartoon characters, memory, reality, and the false belief scenarios. No pair of cartoon characters were displayed consecutively. All experimental trials were new, creative, and different from previous studies in their computerized animation design and parallel structured design in the unexpected location-change task and unexpected content-change task. All the experimental trials were revised repeatedly to meet the need for testing the theory of mind on children. The computerized animation clips were accessed through zenodo online at https://zenodo.org/record/7726825#.ZCVXn-gzaUk (DOI: 10.5281/zenodo.7726825).”

  • It might be helpful to use some Figures to present some key results. Too much information to follow at its current format.

Answer: Three figures were graphed and placed under three sections as Figure 1, 2, and 3. The three tables represented the correct percentage responding to each type of question in 3-year-old and 4-year-old children, correct percentage responding to each type of question in the three participating groups (CA, MA, WS) in the unexpected location-change task, and correct percentage responding to each type of question in the three participating groups (CA, MA, WS) in the unexpected content-change task, respectively. Please see the figures in the text.

Discussion

  • More explanation regarding how the animation-based programs can actually help children with WS or other developmental disabilities to develop better mentalizing skills in reality is needed. 

Answer: In the introduction, a paragraph was added to describe the potential benefits from computer-based technology tools as following: “Computer-based technology tools can enhance social skills in people with developmental disabilities and people with neural disorders through computer-assisted programs, virtual reality, and robotics [14]. In the study by Golan and Baron-Cohen [15], people with Asperger Syndrome (AS) and people with high functioning autism (HFA) were trained using computer-based programs to recognize the contexts of facial and voice emotions. This training was aimed at improving the socio-emotional abilities of these clinical individuals. Participants in the intervention group underwent a 2-hour training every day for 10 to 15 weeks to recognize complex emotions such as insincerity and other mental states. Participants were evaluated before and after the intervention. Each participant was evaluated using several software programs, including the Cambridge Mind-Reading Face-Voice Battery, and psychometric analysis to recognize mental states using formats such as reading the mind in the eyes, voice, and films. Two control groups included people with either of the syndromes AS and HFA who did not receive the interventions and those with typical development. It was hypothesized that the two clinical groups would perform worse before the intervention than typical developers. Moreover, it was hypothesized that, people with AS and people with HFA would improve after taking the training with software programs. These results confirmed the hypotheses. People with AS and people with HFA who received training using interactive, multimedia, and educational software improved in recognizing complex emotions and mental states, proving that computer-based interventions can help people with neurodevelopmental disabilities. Hopefully, people with WS should benefit from such training programs as well.”

Then, the next two paragraphs were added to describe the social skills training program for people with WS [SSTP-WS] and the effect on the population. It was repeated in the following: “Social skills interventions have proven effective for people with neurodevelopmental disorders. Extant research has reported the possibility of using such interventions for people with autism by demonstrating a lack of significant difference between traditional, face-to-face social skills training programs, and behavioral cognitive intervention programs [16]. Fisher and Morin [17] developed interventions for people with WS by using the training programs of UCLA PEERs for Adolescent Programs manual [18], Health and Family Life Education Common Curriculum [19], and Think Social [20]. Before implementing these programs, parental questionnaires were distributed to understand the social skills of people with WS. Next, discussions were held with parents of adults with WS to confirm their social skill problems and to develop specific intervention programs for them (social skills training program for people with WS [SSTP-WS]). Pre- and post-tests of social skills interventions on people with WS were conducted, with effective results observed within two days. Their study demonstrates that SSTP-WS is a promising intervention tool for people with WS.”

“The acceptability, feasibility, and efficacy of this training program were further confirmed by an 8-week-long SSTP with people with WS [21]. Both studies demonstrated effective face-to-face telehealth social skills training in people with WS…..”

These paragraphs clearly stated the effect of computer-based social skill training programs on people with WS. More social skills training programs could be developed in the near future.

Round 2

Reviewer 2 Report

-        The authors’ response breaks down the organization of the introduction for the reviewer, but within those paragraphs, justification for the study remains unclear and unaddressed. As described previously, the introduction includes information about intervention approaches and theories underlying social cognitive difficulties in WS without clear rationale for the inclusion of these components. The manuscript does not detail an intervention study, and in fact does not mention intervention again until the last line of the paper. Moreover, the theories included in this paper are not directly tested or commented on in sufficient detail to make clear how the theories contribute to this paper. While theoretical justification is important, the manner in which it is currently integrated appears to be without clear purpose.

-          The reference to the research question in the response is not actually tested in this paper, as there is no direct test between computerized and non-computerized measures of theory of mind in this study.

-          Claiming “the conclusion is trustworthy” does not enhance the interpretation of the results from this study and is an insufficient response to this reviewer’s concerns.

-          The response to my previous comment about the analysis plan is insufficient and ignores the key issue, namely that the analyses are unclear and unjustified at present, particularly without clear aims and research questions. Importantly, statistical procedures were not clearly defined (e.g., “a binomial statistical test” is a vague description) and the rationale for these approaches do not clearly fall out of the research question. Moreover, clarity on what variables, and how many, are included is unclear. Specifically, was every trial used in analyses? For example, were all 20 false belief questions included in analyses and in the models? Were they averaged for accuracy and included in the model?

-          The rationale for the study, along with hypothesis and predictions, are unclear and untestable the way the study is proposed – what are the questions that motivate the study and why are the variables and approaches used the best method(s) to answering those questions? This was not made clearer in the revision.

Additional concerns

-          The language used in this paper is inappropriate, including the term “mentally retarded” (lines 43, 80) and “Asperger’s”. Even if these terms were previously used in other research, adjustments should be made for the current study.

Reviewer 3 Report

Thank you for addressing my comments and concerns. 

Author Response

We acknowledge and appreciate all the guidance provided.